# Why are some children under 24 months still undernourished in urban and peri-urban Vientiane? A mixed-methods study

Kethmany Ratsavong[1,2]*, Dirk Essink[2], E. Pamela Wright[3], Somphou Sayasone[1], Sengchanh Kounnavong[1], Jacqueline E. W. Broerse[2]

1 Lao Tropical and Public Health Institute (Lao TPHI), Ministry of Health, Sisattanack, Lao People's Democratic Republic, 2 Athena Institute, Vrije University, Amsterdam, the Netherlands, 3 Guelph International Health Consulting, Amsterdam, the Netherlands

¤ Current address: Lao Tropical and Public Health Institute (Lao TPHI), Ministry of Health, Sisattanack, Lao People's Democratic Republic
* kethmany.ratsavong@gmail.com

## Abstract

Child undernutrition remains a significant public health challenge in many low and middle-income countries (LMICs), including Lao PDR, where high levels persist even in urban areas with generally available and accessible food. This study aimed to explore factors underlying the persistently high rates of undernutrition among young children in urban (Saysetha) and peri-urban (Pakgneum) districts of the Vientiane Capital in Lao PDR. A cross-sectional survey employed a sequential explanatory mixed-methods approach, combining a structured questionnaire of 333 mother–child pairs for quantitative analysis with semi-structured interviews of 47 caregivers for qualitative insights. The prevalence of malnutrition among children under 24 months in Vientiane Capital was 27.3% for stunting, 4.2% for wasting, 14.4% for underweight, and 5.11% for overweight. Multiple logistic regression was applied to identify factors associated with malnutrition, while qualitative data were thematically analyzed. The principal findings revealed that, beyond food access, the quality of caregiving and, critically, caregivers' capacity to translate nutrition knowledge into effective practices distinguished well-nourished from undernourished children. Caregivers of better-nourished children obtained health and nutrition information from diverse sources, whereas those of undernourished children relied mainly on health services. In conclusion, strengthening practical nutrition communication in various methods and channels, such as through videos and demonstrations, and enhancing caregivers' ability to apply nutritional knowledge, are central to improving child nutritional outcomes in urban and peri-urban settings in Lao PDR.

**Data availability statement:** All relevant data are within the manuscript and its Supporting information files.

**Funding:** The work was supported by the EU-funded for the Lao Equity through Policy Analysis and Research Networks (LEARN) Program (Number: DCI/SANTI/2014/342-306) and the Medical Committee NetherlandsVietnam in Lao PDR (MCNV Lao PDR). The funders had no role in the study design, data collection and analysis, the decision to publish, or the preparation of the manuscript.

**Competing interests:** The authors have declared that no competing interests exist.

**Abbreviations:** MB, Mother of better well-nourished children; ML, Mother of least well-nourished children; GMB, Grandmother of better well-nourished children; GML, Grandmother of least well-nourished children; FB, Father of better well-nourished children; FL, Father of least well-nourished children.

## Introduction

Maternal and child malnutrition in low- and middle-income countries comprises under-nutrition as well as over-nutrition. Both can affect cognitive development and long-term health and growth outcomes, which are associated with negative effects on individual income as well as economic productivity and social contributions [1]. In recent years, malnutrition in low- and middle-income countries is often a double burden, when both under-nutrition and over-nutrition are present within the same households and communities, especially in Asia [1,2]. Lao People's Democratic Republic (Lao PDR) has high rates of food and nutrition insecurity. Nationally, nearly 36% of children under the age of five are stunted and 27% are underweight in 2015. Malnutrition was estimated to result in 2.4% Growth Domestic Product lost annually (197 million USD) [3].

Suboptimal infant and young child feeding (IYCF) practices, including delayed initiation of breastfeeding, low rate of exclusive breastfeeding, delayed initiation of complementary feeding, low feeding frequency, and low food diversity among children aged 6–23 months, are known to be influenced more by husbands and grandmothers than by health workers [4]. Food insecurity, food restrictions, and taboos applied during pregnancy and for lactating mothers that result in reduced consumption of protein-rich foods are common drivers of maternal and child malnutrition in Lao PDR [5,6]. A recent cross-sectional study on pairs of mothers and children (<6 months of age) in Vientiane City found a high prevalence of inadequate maternal nutrition as well as poor IYCF practices, despite high attendance at antenatal care units [7]. The 2012 Lao Social Indicator Survey noted that one-quarter of the malnourished children resided in urban areas [8].

The persistent high levels of child undernutrition observed in Lao PDR mirror challenges faced across many LMICs, particularly in urbanizing contexts. These data led to the question of why such high levels of malnutrition persist in urban areas where food is available and socioeconomic conditions are good, compared to rural areas. There is, however, little information about mothers' dietary and IYCF practices. To address the undernutrition problem in urban areas, a better understanding of both the extent and its causes are needed. The aims of this study were to explore differences, particularly in child care, that could explain the persistently high level of undernutrition among young children in the urban and peri-urban districts where food is available, accessible, and affordable to most families.

## Methods

### Study area and population

Initially, this study was part of the linear programming approach using the OPTIC-FOOD study to develop food-based recommendations (FBRs) for children under two. Consequently, the first phase of data collection was planned for a sample of 420 mother-child pairs (210 urban and 210 peri-urban) [9].

To represent these settings, we used the list of all districts in Vientiane Capital to randomly select one district representing the urban area (Saysettha) and one

representing the peri-urban area (Pakgeum, more than 60 km from the city center), as they represent the rapid nutritional transition occurring in Vientiane Capital rather than provide national representativeness. First, five villages in Saysettha and two villages in Pakgeum were selected for data collection based on a random selection from the list of villages in the district. The initial expected number of children was based on population records provided by the Lao Statistics Bureau.

However, when these numbers turned out to be underestimated, we expanded the near catchment area of the selected villages. Finally, we include seven villages in the peri-urban area and eight villages in the urban area (Table 1). However, we continued to face challenges, including underestimates of the number of children in the capital, limited time constraints, limited resources, and a high number of incomplete questionnaires due to the unavailability of the caregiver.

We ultimately obtained data from 333 pairs of mothers and children aged under 23 months for final analysis (Table 1). The final number of recruited children was calculated based on the requirements for a cross-sectional study (see Annex 1 in S1 File for details of the sample size calculation). This process ensured that we retained enough statistical power for the subsequent analysis of this study.

## Study design

We carried out a cross-sectional study using a sequential explanatory mixed methods approach. The data were collected in two phases; first, the quantitative data were collected from 23 January2019–17 March 2019, with analysis starting in 01 April 2019. The qualitative data were collected in the second phase between 18–23 December 2019, to explore caregiving behaviors, perceptions, and contextual practices that could help explain the quantitative findings, based on the outcome of the first phase. (See Annex 1 and Annex 1.2: Criteria to select the sample for the qualitative study in S1 File).

## Field procedures

**The quantitative survey. Data collection:** The mother or primary caregiver was asked if they were willing to participate in the study, and signed a consent form before a face-to-face interview. Any caregiver and child who were not

**Table 1. Numbers of children in study villages District.**

|  | Zone | Village | Number of pair mother and child |
|---|---|---|---|
| Pakgeum, N = 145 | Peri-urban | Donsangphai | 43 |
|  |  | Donhai | 36 |
|  |  | Ban phao | 38 |
|  |  | Ban hai | 24 |
|  |  | Somsavard | 6 |
|  |  | Thakokhai | 5 |
|  |  | Nabong | 10 |
| Saysettha, N = 155 | Urban | Vangsai | 10 |
|  |  | Nasangphai | 50 |
|  |  | Nakuay kang | 19 |
|  |  | Haikham | 21 |
|  |  | Nakuay tai | 18 |
|  |  | Nonsagna | 21 |
|  |  | Somsagna | 23 |
|  |  | Chommany | 9 |
| Total |  |  | 333 |

available at home during the survey, or households having a child with a disability or chronic disease, were excluded. Each household was interviewed over three days (Scheme 1).

Dietary data were collected using a 24-hour food recall questionnaire and a multi-day food tally. The tally was recorded by a data collector on Day 1, followed by maternal recording from Day 2 to Day 5. To ensure accuracy in food identification and portion size estimation, a standardized local food photobook was used to guide participants during the interviews. Additionally, food weighing was performed by data collectors using digital kitchen scales accurate to 1 gram.

Anthropometric measurements were taken by maternal and child health nurses from Vientiane Health Office, with an assistant trained as a data collector. To minimize inter-examiner variability, and maximize reliability, among the data collection team, all had three days of training (two days for the questionnaire and one for anthropometry orientation); one more day was used to pretest the form. We used baby-and-mother weighing scales (SECA 383 and SECA 874) accurate to the nearest 50 grams. Children's recumbent length (SECA 416) and the mothers' height (SECA 213) were measured to the nearest 0.1 cm. When children were irritable, they were weighed in the caregiver's arms on a weighing scale with a mother-child weighing function (SECA 874), to a precision of 50 grams. All personnel participated in a standardization workshop. Measurements were performed using a two-person team approach (one measurer and one recorder/verifier) to reduce transcription errors and measurement drift. Standardization was maintained through the use of high-precision SECA equipment across both districts, and measurements were repeated if the child's positioning was unstable.

The data were recorded using an electronic CommCare® application and Samsung Galaxy Tab A. Data were collected from 165 pairs of mothers and children in Pakgeum district and 183 pairs in Saysettha. None of the invited mothers declined to participate in the survey.

The questionnaire included questions on socio-demographic information, food security, IYCF knowledge, and several addressing women's empowerment.

**Data analysis:** We cleaned the data in Excel before importing to Stata 17 for analysis. After deleting incomplete questionnaires, 333 mother-child pairs were included. The World Health Organization (WHO) Anthro Survey Analyzer software was used to analyze the child anthropometric data. The descriptive statistics used proportions or percentages; differences between groups were compared for significance using Chi-square and Fisher exact tests. Variables included in the multiple logistic regression (MLR) models were selected based on theoretical relevance, prior literature, and statistical significance in the bivariate analysis. A manual backward model reduction approach was used to identify the final models.

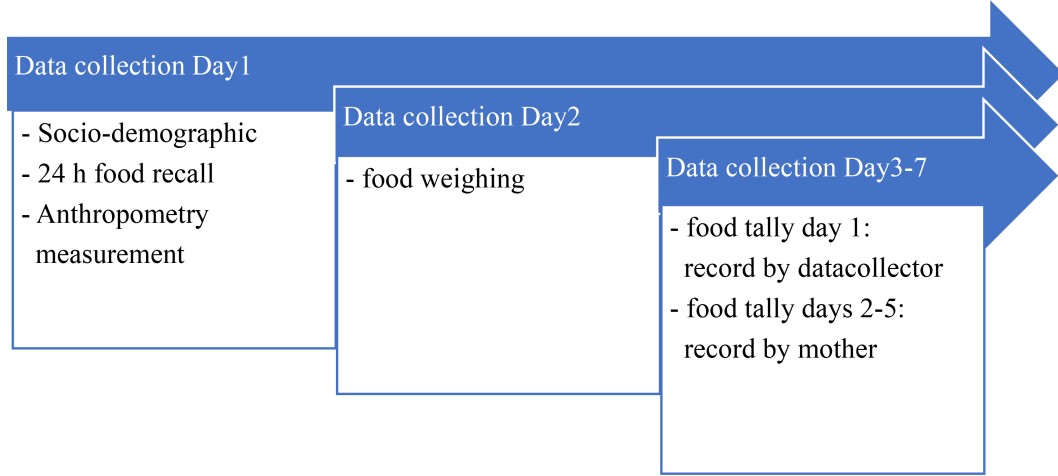

**Scheme 1. Data collection.**

Multiple logistic regression (MLR) was carefully evaluated for potential confounding and multicollinearity. Model assumptions, potential confounding among associated factors, and overall model fit were evaluated using the Hosmer–Lemeshow goodness-of-fit test and Receiver Operating Characteristic (ROC) curve analysis. Confidence intervals and effect sizes for odds ratios are provided to inform the strength and precision and were used to identify factors associated with malnutrition. Differences with a p-value less than 0.05 were considered significant. IYCF and food safety scores were calculated by adding correct answers, calculating the means, and re-grouping the variables in two categories. Those with a value under the mean were considered low scores, and those equal to or above the mean were considered high scores.

   **Definitions.** World Health Organization (WHO), Child Growth Standards [10]:

- **Well-nourished children**: were not stunted, not underweight, and not wasted, nor were they overweight/obese, using anthropometric indicators (Weight fore Age Z-score (WAZ) <-1 to ≥ -2; Length for Age Z-score (LAZ) <-1 to ≥ -2, Weight for Length Z-score (WLZ) <-1 to ≥ -2).

- **Stunted children:** were too short for their age, who had the anthropometric indicator Length-for-Age Z-Score (LAZ) <-2

- **Wasted children:** were too thin for their height, who had the anthropometric indicator Weight-for-Length Z-Score (WLAZ) <–2 SD

-  **Underweight children:** had too low weight for their age, using the anthropometric indicator Weight-for-Age Z-Score (WAZ) <-2 SD

- **Overweight children:** were too heavy for their age, using the anthropometric indicator Weight-for-Age Z-Score (WAZ)>+2 SD for analytical purposes in this study. However, Weight-for-Length Z-score (WHZ/WLZ) is more commonly recommended by WHO for assessing overweight among children under five years.

- **The Body Mass Index** (**BMI):** The Body Mass Index of mothers was calculated using the formula BMI = $kg/m^2$, where kg is the weight in kilograms and $m^2$ is the height in meters squared, then categorized using the Asian classification (11).

-  **Underweight mother:** with BMI <18.5

-  **Normal weight mother** with BMI between 18.5 and 22

-  **Overweight mother** with BMI between 23 and 24

-  **Obese mother** with BMI equal to 25 or above

- **Weight< mean:** The cut off value when the weight of the mother was less than the average weight of all mothers in the study.

-  **Weight>= mean:** The cut off value when the weight of the mother was more than the average weight of all mothers in the study.

-  **Height <mean:** The cut off value when the height of the mother was less than the average height of all mothers in the study.

-  **Height>= mean:** The cut off value when the height of the mother was more than the average height of all mothers in the study.

## Qualitative interviews

   **Study population.** Preliminary socio-demographic and anthropometric data from the survey served to identify households with either well-nourished or the least well-nourished children (Annex 1.1 in S1 File). Further investigation of these households aimed at understanding the high prevalence of undernourishment even in these non-poor areas. For

qualitative data collection, households with well-nourished children were identified as those with children who were not stunted, underweight or wasted, also not overweight/obese, with anthropometric indicators WAZ < −1 to ≥−2; LAZ < −1 to ≥ −2, WLZ < −1 to ≥−2. To select households for interviews, we added the scores on the other surveys, IYCF, and vaccination records. Households with a score in each survey equal to or above half the overall score were considered households with well-nourished children, while those with least well-nourished children were defined as having children with scores WAZ < −2; LAZ < −2, WLZ < −2, and scores on the other records under half of the overall score (Annex 1.2 in S1 File) [10]. We returned to the study villages and randomly selected caregivers in households with well-nourished and least well-nourished children for semi-structured interviews. The target groups in each village included different adults who contributed to child care: four mothers (two of the well- and two of least well-nourished), four fathers (two and two again), and four grandmothers (two and two).

**Data collection.** Only one interviewee was selected per household; those who agreed to participate signed an informed consent form before the face-to-face interview started. Each interview had one interviewer and one note-taker; a recorder enabled transcription of the interviews. When none of the target caregivers were available in a selected household, we moved to another selected household using the same criteria. Finally, 47 caregivers were interviewed (Table 2).

The semi-structured interview guidelines were based on standard IYCF care [11], as well as breastfeeding, vaccination, hygiene practices, and coping strategies, to identify what might influence good IYCF practices by caregivers of well-nourished children. We sought reasons for children having good or poor nutritional status in relation to caregivers' knowledge, beliefs and practices on infant and young child feeding. The interviews lasted an average of 43 minutes (range 28–76 minutes).

**Data management and analysis.** All semi-structured interview data were transcribed verbatim immediately following the interviews to ensure data integrity. The transcripts were analyzed using ATLAS.ti version 8 software. To minimize classification bias, a thematic analysis was conducted following an open-coding approach. Initially, two researchers independently reviewed a subset of transcripts to develop a preliminary codebook. These codes were then categorized into broader themes based on the study guidelines. Discrepancies in the application of codes were resolved through peer debriefing and consensus-building sessions. This iterative process ensured that the final themes accurately represented the lived experiences of the caregivers in both urban and peri-urban settings.

## Ethical considerations

This study was conducted according to the guidelines laid down in the Declaration of Helsinki and all procedures involving research study participants were approved by the National Ethics Committee for Health Research (NECHR), in Lao

**Table 2. The numbers of caregivers interviewed per village.**

| Village | Well-nourished | | | Least well-nourished | | | Total |
|---|---|---|---|---|---|---|---|
| | F | M | GM | F | M | GM | |
| Nasangphai (urban) | 5 | 3 | 3 | 1 | 1 | 0 | 13 |
| Somsagna (urban) | 2 | 2 | 3 | 1 | 2 | 1 | 11 |
| Donhai (Peri-urban) | 2 | 4 | 1 | 2 | 1 | 2 | 12 |
| Donsangphai (Peri-urban) | 2 | 2 | 3 | 2 | 1 | 1 | 11 |
| Total | 11 | 11 | 10 | 6 | 5 | 4 | 47 |

F = Father; M = mother; GM = Grandmother.

PDR (No. 04/NECHR, Vientiane capital 2019). Written informed consent was obtained from all subjects. All participants received an informed consent form prior to the interviews that confirms that the participants are not coerced, persuaded, or induced into the research against their will. The informed consent forms were often explained to the participants before they signed them. In cases of illiteracy, they signed by fingerprint. The data collected was handled anonymously, stored securely, and was only used for scientific purposes.

**Inclusivity in global research.** Additional information regarding the ethical, cultural, and scientific considerations specific to inclusivity in global research is included in the Supporting Information (S5 Checklist)

## Results

The study populations in the urban and peri-urban villages had similar gender and residence proportions; among 333 children, 52.8% were boys; 47.1% resided in peri-urban villages. A few significant differences were found in the sociode-mographic characteristics of the urban and peri-urban villages. The number of children in the household in the peri-urban district was significantly higher than in the urban district, while the household income (HH), interviewee's income, household expenditures, and food expenditures were significantly higher in the urban than in the peri-urban district. More interviewees in the peri-urban district reported higher frequencies of health education and health training than in the urban district. However, only a few households in the peri-urban district had briefly experienced food insecurity. The prevalence of malnutrition of children under two years old was high in both the urban and the peri-urban districts; there was no significant difference between the two. Among all children, 27.3% were stunted, 4.2% were wasted, 14.4% were underweight, and 5.11% were overweight; the remainder were well-nourished. Among mothers, the prevalence of malnutrition was significantly different between the two districts (Table 3).

### Factors associated with child nutritional status

**Well-nourished children.** Certain factors were significantly associated with well-nourished children. Boys were significantly less well-nourished than girls, according to MLR (OR= 0.230, CI 95% [0.119–0.446]). Mothers with less than average height were less likely to have well-nourished children, MLR, (OR= 0.364, CI 95% [0.197–0.674], while overweight mothers were eight times more likely to have well-nourished children MLR, (OR= 8.337, CI 95% [2.404–28.907]). Children living with three or more in the same household were less likely to be well nourished than children in other groups, MLR, (OR= 0.404, CI 95% [0.168–0.973]. Interviewees who had an annual income of more than 1.100.000–3.000.000 LAK (66–180 USD) were more likely to have a well-nourished child, MLR (OR=4.542, CI 95% [1.076–19.166]), (data in Annex 2 and Annex 3.1 in S1 File).

**Stunted and wasted children.** Boys were significantly more likely to be stunted, according to MLR (OR=2.098, CI 95% [1.077–4.088]). Mothers or children with less than average height were significantly more likely to have stunted children or to be stunted [mother's height, (MLR (OR=2.702, CI 95% [1.384–5.277])]; child's height, MLR (OR=11.342, CI 95% [3.214–40.020])]; while overweight mothers were significantly less likely to have a stunted child, MLR (OR= 0.096, CI 95% [0.023–0.400])]. Children aged from 18–23 months were more likely to be stunted than younger children, aged 11–18 months, MLR (OR=3.551, CI 95% [0.976–12.921]); age 18–23 months, MLR (OR=6.348, CI 95% [1.285–31.355]).

Households with three or more other children aged 5–15 years were significantly more likely to have a stunted child MLR (OR=9.278, CI 95% [1.789–48])]. while the child who stays with other caregivers than the mother, such as a grandparent or male household head, was significantly more likely to have malnutrition: grandparent MLR (OR=20.871, CI 95% [2.311–188.526]); male household head, MLR (OR=18.650, CI 95% [1.969–176.632]) compared to mother of the child itself who are not household head, MLR (OR=6.245, CI 95% [1.252–31.162]). No other associations with stunting were found (Annex 2 and Annex 3.2 in S1 File).

The only significant association with wasting was being a boy, MLR (OR= 4.948, CI 95% [1.030–23.767]). (Annex 2 and Annex 3.3 in S1 File).

 

**Table 3. Socio-demographic and nutritional status of mothers and children in peri-urban and urban districts.**

| Socio-demographic & Nutritional status | Pakgeum, Peri-urban, n=162 | | Saysetha, urban, n=171 | | Overall, % (n), n=333 | | P-value[a] |
|---|---|---|---|---|---|---|---|
| | %(mean) | n (SD) | %(mean) | n (SD) | %(mean) | n (SD) | |
| **Gender** | | | | | | | |
| Boys | 54.94 | 89 | 50.88 | 87 | 52.85 | 176 | 0.42[1] |
| Girls | 45.06 | 73 | 49.12 | 84 | 47.15 | 157 | |
| **Age in month** | | | | | | | |
| Mean (SD) | 9.68 | 6 | 10.39 | 6 | 10.06 | 6 | 0.31[1] |
| **Age group** | | | | | | | |
| <6 | 32.10 | 52 | 30.41 | 52 | 31.23 | 104 | 0.73[1] |
| 6-11 | 30.25 | 49 | 27.49 | 47 | 28.83 | 96 | |
| 12-17 | 28.40 | 46 | 29.24 | 50 | 28.83 | 96 | |
| 18-23 | 9.26 | 15 | 12.87 | 22 | 11.11 | 37 | |
| **Number of siblings** | | | | | | | |
| no brother/sister | 19.87 | 30 | **32.75** | 56 | 26.71 | 86 | **0.01[1]** |
| 1 brother/sister | 32.45 | 49 | **33.92** | 58 | 33.23 | 107 | |
| 2 brothers/sisters | 28.48 | 43 | 23.98 | 41 | 26.09 | 84 | |
| >=3 brothers/sisters | 19.21 | 29 | 9.36 | 16 | 13.98 | 45 | |
| **Number of children <5 yrs. in HH** | | | | | | | |
| 0 other ch<5yrs. | 0.00 | 0 | 7.60 | 13 | 3.90 | 13 | |
| 1 other ch<5yrs. | 73.46 | 119 | 78.36 | 134 | 75.98 | 253 | **<0.001[4]** |
| 2 other ch<5yrs. | 19.75 | 32 | 13.45 | 23 | 16.52 | 55 | |
| >=3 other ch<5yrs. | 6.79 | 11 | 0.58 | 1 | 3.60 | 12 | |
| **Number of the children 5–15 yrs. in HH** | | | | | | | |
| 0 other ch 5–15 yrs. | 41.98 | 68 | 46.20 | 79 | 44.14 | 147 | 0.79[1] |
| 1 other ch 5–15 yrs. | 35.80 | 58 | 35.67 | 61 | 35.74 | 119 | |
| 2 other ch 5–15 yrs. | 17.28 | 28 | 14.04 | 24 | 15.62 | 52 | |
| >=3 other ch5–15 yrs. | 4.94 | 8 | 4.09 | 7 | 4.50 | 15 | |
| **Number of adult males aged >15 yrs. in HH** | | | | | | | |
| <=1 Adult male | 51.85 | 84 | 47.37 | 81 | 49.55 | 165 | 0.80[1] |
| 2 Adult males | 25.93 | 42 | 29.24 | 50 | 27.63 | 92 | |
| >=3 Adult males | 14.81 | 24 | 16.96 | 29 | 15.92 | 53 | |
| >=4 Adult males | 7.41 | 12 | 6.43 | 11 | 6.91 | 23 | |
| **Number of adult Female aged >15 yrs. in HH** | | | | | | | |
| <=1 Adult Female | 41.98 | 68 | 41.52 | 71 | 41.74 | 139 | 0.49[1] |
| 2 Adult Females | 38.27 | 62 | 32.16 | 55 | 35.14 | 117 | |
| >=3 Adult Females | 12.35 | 20 | 16.37 | 28 | 14.41 | 48 | |
| >=4 Adult Females | 7.41 | 12 | 9.94 | 17 | 8.71 | 29 | |
| **Vaccine card available in HH** | | | | | | | |
| Yes | 93.83 | 152 | 95.91 | 164 | 94.89 | 316 | 0.39[1] |
| **Illness in the last 2 weeks diarrhea** | | | | | | | |
| Yes | 1.23 | 2 | 0.58 | 1 | 0.9 | 3 | 0.48[1] |
| **Fever** | | | | | | | |
| Yes | 6.79 | 11 | 5.85 | 10 | 6.31 | 21 | 0.72[1] |
| **Cough** | | | | | | | |
| Yes | 8.64 | 14 | 5.26 | 9 | 6.91 | 23 | 0.22[1] |

*(Continued)*

**Table 3.** (Continued)

| Socio-demographic & Nutritional status | Pakgeum, Peri-urban, n=162 | | Saysetha, urban, n=171 | | Overall, % (n), n=333 | | P-value[a] |
|---|---|---|---|---|---|---|---|
| | %(mean) | n (SD) | %(mean) | n (SD) | %(mean) | n (SD) | |
| **Annual HH income** | | | | | | | |
| 50.000.000 | 8.61 | 13 | **17.68** | 29 | 13.33 | 42 | **<0.001[4]** |
| >30.000.000-50.000.000 | 12.58 | 19 | **23.78** | 39 | 18.41 | 58 | |
| >10.000.000-30.000.000 | **23.18** | 35 | **28.05** | 46 | 25.71 | 81 | |
| >3.000.000-10.000.000 | **45.03** | 68 | 25.00 | 41 | 34.60 | 109 | |
| 1.100.000 - 3.000.000 | **5.96** | 9 | 3.05 | 5 | 4.44 | 14 | |
| <1100000 | **4.64** | 7 | 2.44 | 4 | 3.49 | 11 | |
| **Annual interviewee Income** | | | | | | | |
| >50.000.000 | 5.30 | 8 | 2.44 | 4 | 3.81 | 12 | **0.003[4]** |
| >30.000.000-50.000.000 | 3.97 | 6 | **13.41** | 22 | 8.89 | 28 | |
| >10.000.000-30.000.000 | **25.17** | 38 | 24.39 | 40 | 24.76 | 78 | |
| >3.000.000-10.000.000 | **31.13** | 47 | 28.05 | 46 | 29.52 | 93 | |
| 1.100.000 - 3.000.000 | **11.92** | 18 | 3.66 | 6 | 7.62 | 24 | |
| <1100000 | 22.52 | 34 | 28.05 | 46 | 25.4 | 80 | |
| **Who manage the income** | | | | | | | |
| Head of HH | **16.25** | 26 | 6.43 | 11 | 11.18 | 37 | **<0.001[4]** |
| Father of the child | 3.75 | 6 | 1.17 | 2 | 2.42 | 8 | |
| Mother of the child | 53.13 | 85 | **77.78** | 133 | 65.86 | 218 | |
| Other family members | **26.88** | 43 | 14.62 | 25 | 20.54 | 68 | |
| **Primary caregiver** | | | | | | | |
| Female (household head) | 14.81 | 24 | 7.02 | 12 | 10.81 | 36 | **0.02[4]** |
| Female (non-head of household) | 69.75 | 113 | **81.87** | 140 | 75.98 | 253 | |
| Male household head | 5.56 | 9 | 2.34 | 4 | 3.90 | 13 | |
| Grandparents | 6.79 | 11 | 8.19 | 14 | 7.51 | 25 | |
| Another member in HH | 3.09 | 5 | 0.58 | 1 | 1.80 | 6 | |
| **HFIAS*** | | | | | | | |
| Yes | 7.01 | 11 | 0.00 | 0.00 | 3.36 | 11 | **<0.001[4]** |
| No | 92.99 | 146 | 100 | 170 | 96.64 | 316 | |
| **Market in the village** | | | | | | | |
| Yes | 32.72 | 53 | 45.03 | 77 | **39.04** | 130 | **0.02[1]** |
| No | 67.28 | 109 | 54.97 | 94 | 60.96 | 203 | |
| **The expenditure in the last 30 days (Kip)** | | | | | | | |
| Mean food expenditure | 1540369 | 1925263 | 3147313 | 15400000 | 2383275 | 11200000 | **0.0001[3]** |
| Mean household expenditure | 1523058 | 1907229 | **3080276** | 14900000 | 2356306 | 11000000 | **<0.001[3]** |
| **Child nutritional status** | | | | | | | |
| Child average weight, Mean (SD) | 7.91 | 2 | 7.96 | 2 | 7.94 | 2 | 0.86[2] |
| The proportion of weight>= mean | 46.3 | 75 | 47.95 | 82 | 47.15 | 157 | 0.76[1] |
| The proportion of weight <mean | 53.7 | 87 | 52.05 | 89 | 52.85 | 176 | |
| **Child average height, Mean (SD)** | 67.93 | 8 | 69.60 | 9 | 68.79 | 8 | 0.09[3] |
| The proportion of height>= mean | 48.15 | 78 | 53.22 | 91 | 50.75 | 169 | 0.36[1] |
| The proportion of height<mean | 51.85 | 84 | 46.78 | 80 | 49.25 | 164 | |
| Healthy | 58.02 | 94 | 67.25 | 115 | 62.76 | 209 | 0.08[1] |
| Stunted | 29.63 | 48 | 24.56 | 42 | 27.03 | 90 | 0.30[1] |
| Wasted | 3.70 | 6 | 4.68 | 8 | 4.2 | 14 | 0.66[1] |

*(Continued)*

**Table 3.** (Continued)

| Socio-demographic & Nutritional status | Pakgeum, Peri-urban, n=162 | | Saysetha, urban, n=171 | | Overall, % (n), n=333 | | P-value[a] |
|---|---|---|---|---|---|---|---|
| | %(mean) | n (SD) | %(mean) | n (SD) | %(mean) | n (SD) | |
| Underweight | 12.96 | 21 | 15.79 | 27 | 14.41 | 48 | 0.46[1] |
| Overweight | 7.41 | 12 | 2.92 | 5 | 5.11 | 17 | 0.06[1] |
| **Mother nutritional status** | | | | | | | |
| Mean weight of mother (Kg), (SD) | 54.97 | 10 | **55.17** | 11 | 55.10 | 10 | 0.79[3] |
| The proportion of weight<mean | **45.39** | 69 | 43.31 | 68 | 44.34 | 137 | 0.71[1] |
| The proportion of weight>= mean | 54.61 | 83 | **56.69** | 89 | 55.66 | 172 | |
| Mean height of mother (meters) | 1.52 | 0.05 | **1.53** | 0.05 | 1.53 | 0 | 0.20[3] |
| The proportion of height>=mean | **50.00** | 81 | 56.73 | 97 | 53.45 | 178 | 0.22[1] |
| The proportion of height<mean | **50.00** | 81 | 43.27 | 74 | 46.55 | 155 | |
| Normal weight | 43.42 | 66 | **43.95** | 69 | 43.69 | 135 | **0.01[1]** |
| Underweight | 5.26 | 8 | **12.1** | 19 | 8.74 | 27 | |
| Overweight | **21.05** | 32 | 8.92 | 14 | 14.89 | 46 | |
| Obese | 30.26 | 46 | **35.03** | 55 | 32.69 | 101 | |
| **Mother's age group** | | | | | | | |
| <=24 | 32.72 | 53 | **30.41** | 52 | 31.53 | 105 | **0.36[1]** |
| 25-35 | 53.09 | 86 | **59.65** | 102 | 56.46 | 188 | |
| >35 | 14.20 | 23 | **9.94** | 17 | 12.01 | 40 | |
| **Mother Education** | | | | | | | |
| >College, professional or higher | 9.94 | 16 | **34.91** | 59 | 22.73 | 75 | **<0.001[1]** |
| <=High school | 90.06 | 145 | 65.09 | 110 | 77.27 | 255 | |
| **IYCF Knowledge score** | | | | | | | |
| Mean (SD) | 4.65 | 0.93 | 10.29 | 3.39 | 7.54 | 3.78 | **<0.001** |
| IYCF score>=mean | **0.00** | 0.00 | 77.19 | 132.00 | 39.64 | 132.00 | **<0.001[4]** |
| IYCF score<mean | 100.00 | 162.00 | **22.81** | 39.00 | 60.36 | 201.00 | |
| **Food safety score** | | | | | | | |
| Mean | 0.44 | 0 | **0.46** | 1 | 0.45 | 0 | **0.75[3]** |
| Food safety score>= mean | **55.56** | 90 | 53.8 | 92 | 54.65 | 182 | **0.83[1]** |
| IYCF score<mean | 44.44 | 72 | **46.2** | 79 | 45.35 | 151 | |
| **Ever received health training or health education** | | | | | | | |
| Yes | 91.98 | 149 | **97.66** | 167 | 94.89 | 316 | **0.02[4]** |
| **Training on Antenatal Care** | | | | | | | |
| Yes | **4.32** | 8 | 0.58 | 1 | 2.40 | 8 | **0.03[4]** |
| **Training on how to take care of the child after birth** | | | | | | | |
| Yes | **4.94** | 7 | 0.58 | 1 | 2.70 | 9 | **0.02[4]** |
| **Training on vaccination** | | | | | | | |
| Yes | **5.56** | 9 | 1.17 | 2 | 3.30 | 11 | **0.03[4]** |
| **Training on childcare** | | | | | | | |
| Yes | **5.56** | 9 | 1.17 | 2 | 3.30 | 11 | **0.03[4]** |

Statistic used: [1] Chi2 square, [2] t-test, [3] Wilcoxon rank test, [4] Fisher exact test, * HFIAS= Household Food Insecurity Access Scale.

**Underweight and overweight children.** Boys were significantly more likely to be underweight than girls, MLR (OR=5.097, CI 95% [1.795–14.471]). Children with a weight less than the average among the study population were more likely to be undernourished also by international standards, MLR (OR=25.885, CI 95% [5.236–127.961]. Underweight was also more likely in the children from 18–23 months, MLR (OR=14.635 [2.121–100.968] than in younger children, 6–11 months, MLR (OR=0.228 [0.066–0.786]. The households reporting no vaccination card also had higher probability of an underweight child, MLR (OR= 8.932, CI 95% [1.105–72.222]).

Mothers whose weight and height were less than average were more likely to have an underweight child: low weight, MLR (OR=10.991, CI 95% [2.998–40.289]); low height, MLR (OR=4.723, CI 95% [1.709–13.051]).

Those with low to medium household incomes were more likely to have underweight children: annual HH income (>3.000.000–10.000.000), MLR (OR=16.605, CI 95% [1.659–166.149]). Furthermore, households with two or more daughters/sons were significantly more likely to have underweight children, and the probability increased with the number of children in the family: two siblings, MLR (OR=3.666, CI 95% [1.07012.559]); three or more siblings, MLR (OR=**5.017**, CI 95% [1.281–19.649]).

Perhaps unexpectedly, children in households with a lower-than-average IYCF score were less likely to be under-weight: IYCF score<mean MLR (OR=0.179, CI 95% [0.06–0.526]). No other associations with underweight were found (Annex 2 and Annex 3.4 in S1 File).

Children whose height was below average were less likely to be overweight than other groups, MLR (OR= 0.138, CI 95% [0.029–0.656]). Children who reside in an area where is no market were more likely to be overweight, MLR (OR= 3.523, CI 95% [1.065–11.653]); No other associations with overweight were found (Annex 2 and Annex 3.5 in S1 File).

## Deeper insights into care and feeding of well-nourished & undernourished children

From the quantitative data, it was clear that in both urban and rural districts, high levels of undernutrition were common. To gain a deeper understanding of the care and feeding practices among households with well-nourished and least well-nourished children, we interviewed caregivers from both groups. Analysis revealed a number of themes that provided information about aspects of care that could contribute to whether the child is well-nourished or not.

**Which foods do caregivers consider good for the child?.** Both groups of caregivers mentioned that food from natural sources is good for the child's health because it has fewer or no chemicals; they mentioned river fish and their own garden vegetables as preferable to commercial products. Caregivers of well-nourished children recommended correct nutritious foods, giving examples of vitamins and proteins. Caregivers of undernourished children gave similar recommendations but could name fewer good foods, and did not mention vitamins. They also made incorrect recommendations such as giving sticky rice to make the child feel full, and using instant and processed foods.

*"The food that is good for the child could be fish or egg. If it is natural fish is it is good for health but the fish raised in floating basket is high in chemicals. I think egg is good because it has high protein. Fruit and vegetable is good for the child because child can have easy defecation. I think is good vegetables are the green leafy ones such as spinach, morning glory, ground ivy, bok choy, because they come from a natural garden with no chemicals. Good fruits include oranges, which have vitamin C, and apples are also good" (Father of well-nourished child, FB1)*

*"The food that is good for the child is rice, fruit, vegetables…sticky rice is better than white rice because if we let the child eat white rice, they will be hungry again very soon…the sticky rice will make the child full for a long time. Fruit that good is apple, banana, orange, papaya…and vegetables like bok choy"* **(Father of least well-nourished child, FL1)**

**Which foods are not good for children?.** The two groups of caregivers identified similar foods as not good for the child, including fermented, salty, spicy or sour foods, and unhygienic, spoiled, alcohol-containing, and chemical foods, with most of the concern related to belly pain and diarrhea in children.

*"I never give him salty food, and fermented food, it is not good for the child's stomach…the child will have diarrhea"* *(Grandmother of well-nourished child, GMB1)*

*"The ready-to-eat food that we buy from the market is not recommended to give to the child because it might not be clean and has been prepared for a long time already…Spicy or sour food …will cause belly pain and diarrhea, such as fermented vegetables, or soup that adults eat with chili as fish soup, chicken soup…"* *(Father of the least well-nourished child, FL4)*

**Sources of information related to child care.** More mothers of well-nourished children indicated that they received information from social media, mass media (television, radio, newspaper), health campaigns, and relatives. Caregivers of the least well-nourished children said they obtained information mostly from the pink book (given during antenatal care), their doctor, the village authorities, and peers during ante-natal care; only a few reported using social media to obtain information.

*"I heard from the senior women who have experience with child care, such as relatives and friends, also I saw on YouTube where mothers share their experience…. I searched the keyword 'food for children'…I also searched on Google using Thai language … and I listen to the Lao radio 9.7, they talk about children's food."* *(Mother of well-nourished child, MB8)*

*"I know which food is good or not good because the doctor from hospital gave the recommendation to me…it is also in the pink book."* *(Mother of least well-nourished child, ML2)*

**Recommended foods versus what is given to the child.** The caregivers of well-nourished children prepared food that they thought is easy for the child to eat and digest, like soup, white rice, or rice porridge. There was consistency between the recommended food and their own practice to provide that recommended food to the child in reality. They often mentioned soup, which they made with pork bone, vegetables, and meat. Soup can make rice softer and easier to chew and swallow. They also explained that soup is given using a spoon, avoiding unhygienic use of hands, unlike sticky rice, which requires eating with hands and is typically accompanied by dry foods that may be associated with constipation and difficult for young children to chew. Snacks were in the form of vegetables and fruit. The caregivers of well-nourished children tried to give food that they think is good for the child, whether the child likes it or not, even if the child doesn't like to eat it. There was consistency between what the caregivers recommended and what they actually gave to the child.

The caregivers of the least well-nourished children recommended similar foods. However, they did not translate their recommendations into action. They prepared types of food that are convenient for caregivers to bring and keep for a long time, like instant noodles and processed foods. The dried cooked food has only meat and does not provide food diversity; it is usually eaten with sticky rice and without vegetables. When the child eats less, these caregivers let the child decide. When the child refuses to eat, they give other food that fills them, such as extra snacks or milk.

*"The child has 3 meals per day, 2-3 snacks (cannot count), sometimes take banana, sometimes take wheat snack…he takes fried dry meat with sticky rice and he may have fruit or steamed corn as morning snacks, I usually give apple or banana. In the afternoon, I give pork simmered with sticky rice, I chew the pork for him. I also made soup of spinach or ground ivy to help with excretion. Food that is good for the child, I particularly cook vegetables. He has anemia, I give morning glory, spinach, ground ivy, it is good for his intestinal health, the fruit is also good but it is talking money (laughs)…the good fruit is avocado, it will make the child intelligent, apple will make the child healthy, is also good, I give him because he like to eat…fish is also good for him. For milk, I give Lactogen to him, however, no milk is as good as breastmilk…But I am still sick and cannot provide breastfeeding to him."* *(Mother of well-nourished child, MB8)*

*"Yesterday I gave her milk in the morning, not a meal…because I am busy and need to go out to work. …my daughter stays with her grandmother, in the afternoon she takes milk, rice, soy sauce with dry meat…I give it according to the food we have in the house. In the evening, the child takes milk and rice with grilled pork, we cooked and eat together… Yesterday the child did not have any vegetables at all… For fruit, the child has oranges, and boxed 25% orange juice, because she cries to take this drink…. Food I think it is good for the child is fruit, meat, fish, eggs and bananas, because they help for excretion. Vegetables like bok choy are also good for excretion. Fish, pork, and beef are also good especially natural fish…it will increase the child's growth and development."(Father of the least well-nourished child, FL7)*

**Roles of different family members in child care and feeding. Mothers** generally had the main control over the feeding of their children. In families with well-nourished children, the mothers purchased and directed the child's consumption of food. In families with undernourished children, the mothers were often overburdened with tasks, and many were working outside the home. A grandmother might be taking care of two or three small children, leading to a decrease in the quality of care for each child.

*"I take care of him, …give food…bring him to walk around and bring him back to sleep…I am at home doing housework but I do not raise animals, only do child care and housework when the child is sleeping. "(Mother of well-nourished child, MB9)*

*"I give food, milk, shower, and wash clothes…I always stay with her, I am doing everything in the house because I am alone with my child, my husband goes to work, my mum, my sister also go to work…I do cooking, dishes and clothes washing for everyone in the house…", (Mother of least well-nourished child, ML3)*

**Grandmothers** often support child feeding, especially in the urban district where mothers work outside the home. Grandmothers have power when they are responsible for food management and buy food for the whole family. This point was similar between the two groups of caregivers.

*"I help his mother to feed him. His mother decides what food to give to him, because she goes to market to buy the food." (Grandmother of well-nourished child, GMB14)*

*"I am cooking for him, buy food and decide about food for the whole family because his parents work in Thailand, only grandparents stay with him." (Grandmother of well-nourished child, GMB8)*

*"The grandmother decides what food to give the child, because she cooks the food." (Mother of the least well-nourished child, ML10)*

**Fathers** usually work outside the home, providing income support; they play with the child when they return home, while the mother prepares food for the family. Most fathers were not responsible for the caregiving and had less knowledge about the child. This situation was similar for both groups of families.

*"My role for the child is cleaning; cooking is also done by me. Her mother usually doesn't have time because she works in the hospital …most of the time the child stays with me." (Father of well-nourished child, FB1)*

*" I take care of the family, but I have little time to take care of my child. ……I am not a woman, mostly it is my wife, the child's mother, who takes care of her." (Father of the least well-nourished child, FL1)*

**When to start complementary feeding.** Among the 27 well-nourished children, 23 received complementary foods at the appropriate age of 6 months or a little later; three started the complementary food early. Among the least well-nourished children, 5 out of 17 started complementary food early, just after birth, by one month of age, or before 6 months.

**The timing of feeding the child.**  Mothers of well-nourished children indicated that giving food to the child at the same time every day encouraged the child to eat. In contrast, the caregivers of the undernourished children gave food when the child cried, or occasionally skipped meals.

*"I give him food at 6-7 PM every day, at that time if I am not feeding him, he will ask for food, sometimes he brings his bottle to me when he wants to drink water or takes my hand to the sticky rice basket, when he is hungry." (Grandmother of well-nourished child, GMB1)*

*"Mostly I don't know, sometimes the child is not eating at the same time, sometimes the child eats breakfast and lunch, sometimes no breakfast, some days two meals, some days three meals." (Father of undernourished child, FL1)*

**Number and quality of meals/snacks.**  Caregivers of well-nourished children provided three meals per day plus healthy snacks. Among the undernourished children, more received one or two meals per day but more snacks, 4–6 times per day, and more often food with low nutrient value, such as plain rice with salt, bread or cake.

**Coping strategies when the child gets sick.**  Some caregivers respond by obtaining vitamins from a doctor or pharmacy, or by changing the menu, while others just follow the child's health status closely. Caregivers of well-nourished children often encouraged their children to eat more, such as playing with them, taking them out to walk around or changing the food. Caregivers of the least well-nourished children may use such strategies, but often did nothing when the child ate less or did not want to eat.

*"I play with him, hold him up and down…. When he is in a good mood, I will provide food." (Father of well-nourished child, FB4)*

*"She does not eat well, 4-5 portions are enough for her. If she gets sick, I do nothing to increase her food intake." (Father of least well-nourished child, FL6)*

**Child's food preferences.**  Both types of caregivers provided the food that was available in the house and/or the food that they cooked in the same pot for the family to eat together. Caregivers of the undernourished children more often gave the child what he or she liked to eat whether or not that was healthy food.

*"I usually give her dried meat because she only likes to eat dried meat." (Mother of least well-nourished child, ML1)*

**Description by caregivers of a healthy and an unhealthy child.**  Both groups provided similar descriptions of healthy and unhealthy children, using five categories: disease, emotional behaviors, eating behaviors, developmental milestones, and the body composition compared to other children of the same age (Annex 4 in S1 File)

**Comparison of urban and peri-urban contexts.**  Differences: Children in the peri-urban district were given complementary food earlier, and were often given less than three meals per day, filling up on snacks. The mother's role in the family was mainly child care, often on her own with little contribution from other family members and one mother caring for a few children while others go out to work.

Similarities: Food taboos and having to use the hotbed were two traditional practices common to all mothers, directly after giving birth. In the peri-urban areas, the mothers were more concerned about the negative effects of not following these traditions; they were stricter in terms of food intake and used the hotbed longer than did mothers in the urban villages.

**Source of information related to child care.**  Accessing the right source of information through various ways and channels also plays a role in improving child nutrition, including health information disseminated by hospitals, antenatal care clinics, newspapers, radio, television, YouTube, Facebook and peer mothers during waiting for the antenatal service,

along with advice from (grand)mothers, relatives and friends who come to visit after the birth. Having several sources of nutrition and health information can have both positive and negative effects on the child's nutrition; the positive effect is that the well-nourished child's caregivers reported obtaining information more from social media after actively searching, and from sources such as television, radio, newspaper, health campaigns, their relatives. Potential negative effects could come for example from advertising, especially related to breastfeeding substitutes and vitamin supplements that influence mothers to buy, and which are usually not as healthy for the child. The caregivers of the least well-nourished children received more information from the pink book, doctor, village authority, and peer mothers, and much less from social media. They said:

> *"I learned from older women with experience on caring for children, such as relatives, and friends, and from watching YouTube where mothers share their experience….I search the Internet using as keyword, food for children…I also search on Google using Thai language … and listen to the Lao radio 9.7, they talk about children's food" (Mother of a well-nourished child, MB8)*

> *"I know which food is good or not good because the doctor from hospital give the recommendation to me…I also read in the pink book" (Mother of a least well-nourished child, ML2)*

## Discussion

This is the first study in Lao PDR to focus on the nutritional status of children aged under 24 months in urban settings, and provides valuable insights applicable to similar urbanizing contexts across LMICs, where high levels of malnutrition persist despite apparent food availability. The study aimed to explore differences in childcare that might explain the persistently high rates of undernutrition among young children in urban and peri-urban districts where food is generally available, accessible, and affordable to most families. Although the level of undernutrition was notably high in the study areas, few differences were observed between the urban and peri-urban districts; in both settings, food security was not a concern, and food was accessible and affordable for nearly all of the study population.

We first describe the prevalence of malnutrition among young children in Vientiane Capital, the urban and peri-urban areas we studied, where the prevalence was higher compared to that reported for children under five from the Lao Social Indicator Survey II in 2017. According to the WHO cut-off values for public health significance, the prevalence of malnutrition we found would be classified as acceptable for wasting, and of medium prevalence for both underweight and stunting [12]. There were a few overweight children, which is different from reports in neighboring Thailand, where higher rates of overnutrition were reported in urban settings [12]. Studies in China and Vietnam have found that the average weight and height of urban children were greater than peri-urban and rural children [13]. Although our data suggest a tendency in that direction, there were no significant differences between the two environments in the weights or heights of the children. One difference between those studies and ours is that they were systematic reviews looking at population-representative data, while we investigated a smaller purposive sample in the most urbanized part of the country.

The number of boys and girls were not significantly different between the two study sites, but girls were found to be significantly better nourished than boys. The boys were significantly less likely to be healthy and more likely to be stunted, wasted or underweight. We have previously reported similar results on the relation between children's gender and nutritional status [14–16]. Younger children (6–12 months) were less likely to be underweight compared to those from 11–23 months, as reported in Thailand [13]. Possibly the younger children are still breastfed whereas the older ones still need nutrient-dense food but are dependent upon caregivers, who may have started to give the same food as eaten by other family members, especially sticky rice and snacks. The older children also move around more and have a greater chance of picking up an infection, although our results did not demonstrate any direct effect of recent illness. However, children

in households lacking a vaccine card were more often underweight, which suggests a role for illness in their nutritional status.

The maternal nutrition status was classified using Asian categories [11]. The prevalence of maternal underweight was low, but overweight and obesity were highly prevalent, which suggests a serious public health problem. However, because we studied children under two years, some mothers had recently given birth and had not reduced their weight, so these data should be interpreted with caution. The double burden of malnutrition was found, with relatively high prevalence of both under- and over-nourished status in mothers. Maternal nutritional status is related to the nutritional status of their children. Mothers with weight or height less than the average were significantly less likely to have well-nourished children and more likely to have stunted and underweight children, while children whose mothers' weight was less than average were more likely to be underweight. The overweight mothers were significantly less likely to have stunted children, which is similar to results reported from Thailand [12]. We also found that for children of 12–24 months, caregivers started to give food from the same pot as used by adults. This means the child is consuming the same ingredients as the adults, which may explain why mothers with good or above-average nutritional status often have children with similar nutritional status. Since sticky rice is the common starchy staple in Lao [17], and is eaten by hand, children typically eat it either with their own hands or those of their caregivers. This practice may contribute to constipation [18] and increase the risk of infection, especially in contexts where water and sanitation conditions are poor [19].

We found that the mother's nutritional status influenced the child's weight and height. One theory suggests that children's height can be predicted using the average of the parents' heights. A study in Indonesia found that genetic factors from both father and mother influenced stunting [20]. Many studies have found, as we did, that at an early age, boys were more undernourished than girls. In a context of insufficient nutrient intake, such as in households with low socio-economic status, boys, with a higher nutrient requirement than girls, may more easily become undernourished [14–16]. The household composition may influence the nutritional status of young children [21]. Higher numbers of children from one or more mothers in one household can increase the burden of the caregiver and may reduce the quality of care for each child, especially the younger ones requiring highly nutritious food.

### Caregivers' knowledge and practice

We found that the levels of education and knowledge scores for both IYCF and food safety were higher in the urban district than in the peri-urban, even though the number of persons who had been trained on health were greater in the peri-urban area. Many health interventions in Lao PDR are targeted to people in rural areas, but their knowledge scores were still lower than those in the urban district where the training was not provided. Children of caregivers who had low IYCF scores were significantly less likely to be underweight than were children of those with high IYCF scores This apparent discrepancy has been reported by others, who demonstrated that interventions to increase the parents' knowledge were not necessarily associated with improved feeding practices or nutritional status of the child [22], because there were barriers to put the knowledge into practice [23–26]. Caregivers of well-nourished children often obtained information from a range of sources, giving them a broader range of knowledge and ideas about how to put it into practice. IYCF scores only check the caregivers' knowledge, but we could see from the interviews that while knowledge was similar in both groups, practice was different, and the caregivers of well-nourished children could translate their knowledge into practice better than those of the undernourished children.

The caregivers with better practices had children with better nutritional status. In Lao PDR, most official healthcare information comes from the Information Education Communication (IEC) materials in the media or from health professionals, which does not always provide clear guidance on how to apply the information. To effectively change behaviors, greater efforts and clear reference points are needed to positively influence attitudes and ultimately achieve sustained behavioral and physical changes [26,27]. Such efforts are usually made in rural areas, which have priority for nutrition interventions. In behavior change communication, an effective result can be achieved when the population has not only

 

information but also opportunity, ability, and motivation (motivation here refers to attitude related to healthy food) to use that information, based on the local food systems and food environment [28]. At least one of these was missing among the caregivers of the least nourished children. Some caregivers had motivation to see their child healthy, but could not afford the milk or fruit that they think is good for the child, or could only give what is available in their house, while others did nothing, apparently lacking the motivation to make strong efforts to encourage the child to take nutritious food when they refuse or eat less. Other mothers are away all day at work and have little opportunity to buy fresh food, so they chose processed or prepared foods. Finally, some mothers do not know how to implement the knowledge they have, how to convince their child to eat well or how to manage when the child is ill. All of these are similar to findings in Indonesia, that some mothers made fewer efforts to give nutritious food when their child was difficult to feed. Even when mothers had knowledge and knew its importance, there were still barriers to incorporating the knowledge in their practice [26,28,29].

## Source of information on child nutrition

The qualitative results revealed that the caregivers of better-nourished children obtained health and nutrition information from multiple sources, including searching by themselves on social media/internet, including Google, Facebook and YouTube. Such sources provide many demonstrations, teaching both content and application of knowledge on child care and feeding and showing the results. These caregivers were already motivated to take time and look for information; the active type of presentation may further motivate them to use the information to achieve the same results. In contrast, the majority of the caregivers of the undernourished children received health and nutrition information from limited sources, such as reading the pink book on peri-natal care, listening during ante-natal care visits, and from peers, all of which may not motivate the caregiver to follow the recommendations or show them how to do it. Previous studies in Lao PDR found that the communication between mothers and health professionals during ante-natal care visits had many deficiencies and that healthcare providers needed to improve their own knowledge on nutrition and its practice, to integrate into their medical plans [30,31]. Similarly, in Burkina Faso, cooking demonstrations were recommended, because mothers with adequate IYCF knowledge still had inappropriate feeding practice [25], while other studies found that a combined strategy to communicate to mothers had greater impact than using only one strategy [32–34].

The food environment and the household composition can influence everyone's health and nutritional status. When young children live in a household with several other children, the quantity and diversity of food they are given may be limited. Parents experiencing food insecurity reported mainly eating rice and vegetables from their own gardens, which could lead to overconsumption of carbohydrates. In Lao culture, older children should ensure that younger siblings have enough to eat. Parents whose child under five years was a third or higher child would have gained experience and may enjoy a better economic situation than when the first child was born.

## The workload of the caregiver

The household workload, especially of the mother, also plays a role. We suspect that when mothers work long hours, outside the house or when care is provided by a or caregiver such as grandmother looking after several grandchildren, they are more likely to select the ready-to-eat foods [35,36]. In this case, even if mothers have knowledge on good foods, they have no time to cook or may have to delegate care to someone who lacks the correct knowledge, this align with our results found that the child who stay with other caregivers, such as grandparent or male household heads, was more likely to become stunted compared to a child staying with the mother. The knowledge and role of the adult caregivers in the household who contribute to the care practice of the child might be an important factor to improve the nutrition and health status [37]. We did not focus on this issue, but it would be a point for future research. We found that grandmothers might believe that giving any food that makes the child full is acceptable, which could also explain finding undernourished children even when mothers have sufficient knowledge on child nutrition.

An important driver for mothers to give their child healthy food was a good attitude that can increase motivation and improve practice and commitment [26,38,39]. Obtaining ingredients and preparing healthy food takes time and may cost more, especially compared to buying ready-to-eat foods that are easily available. There appeared to be a difference in motivation between the caregivers of well-nourished children and those of undernourished children, but that difference is not explained by our data. The former actively sought information about good child care from a wide range of sources, and applied what they learned, while the latter did that to a much lesser extent, and if the child seemed ill, sought care from the health services. Although we did ask the mothers whether they felt empowered or not, there was no difference between the two groups; both also stated that they were responsible to choose foods for their children. To have a child with good nutritional status, mothers might require motivation, a positive attitude and good knowledge, with good environmental and social support. Further study related to this question is recommended.

## Study strengths and limitations

The main limitation of our study was the lack of recent, disaggregated data on the prevalence of undernutrition specifically among children under 24 months in Vientiane Capital; thus, we utilized the prevalence for children under five from LSIS III as a proxy for sample size calculations [40]. While logistical constraints limited the study to two districts (Saysettha and Pakgeum), these were randomly selected to represent the distinct dynamics of urban and peri-urban environments. We acknowledge the challenges in obtaining precise census data for this specific age group, which we mitigated by expanding our cluster selection to 15 villages. Therefore, the findings may not be fully generalizable to all urban settings in Lao PDR. Furthermore, expansion of village catchment areas due to underestimation of the eligible population may have introduced some selection bias, However, the final recruitment (n = 333) exceeded the statistical requirement by 46% (Annex 1 in S1 File), ensuring the study maintained robust statistical power (>80%) for both the bivariate and multivariate analyses.

In addition, overweight was operationally classified using Weight-for-Age Z-score (WAZ)> +2 SD for analytical purposes, whereas WHO commonly recommends Weight-for-Length Z-score (WHZ/WLZ) for assessing overweight among children under five years.

A significant strength of this study is the use of a 7-day food tally combined with a 24-hour recall, which captured a more stable representation of habitual dietary intake compared to a single-day snapshot. However, we acknowledge that the use of these tools introduces potential recall bias, as mothers may have difficulty accurately remembering the exact frequency and portion sizes of all foods consumed which could have in measurement and reporting bias. Food photobooks, interviewer guidance, and standardized data collection procedures were used to enhance the caregiver recall and improve the accuracy of the dietary reporting.

Second, social desirability bias may have occurred, wherein caregivers might over-report 'healthy' behaviours, such as breastfeeding or vegetable consumption and under-report 'unhealthy' behaviours, such as the provision of ultra-processed snacks, in order to align with perceived expectations from health authorities. To mitigate these risks, we utilized trained enumerators who employed non-judgmental probing techniques and used standardized food photobooks and digital scales to assist in more objective portion estimation.

A further possible limitation is the approximately nine-month interval between the quantitative and qualitative phases of data collection. Although both phases were conducted during the dry season in Lao PDR, seasonal variation in the availability of certain foods, particularly fruits and vegetables, may still have influenced dietary recall and feeding practices. However, because the study was conducted in urban and peri-urban areas where market access and food availability are relatively stable throughout the year, the seasonal effect should not be pronounced. In addition, the qualitative component mainly explored long-term caregiving practices, beliefs, and knowledge application, which are likely to be more stable over time than short-term seasonal food availability.

In the qualitative phase, we interviewed only one primary caregiver per household. In reality, child-rearing in Lao PDR is often shared among multiple family members, whose varied practices may not have been fully captured. We also did not

collect data on certain biological factors, such as birth weight, birth order, or birth spacing, which are known to influence child growth. However, by utilizing a sequential explanatory mixed-methods approach, we were able to capture unique socio-behavioural aspects of childcare and feeding such as the role of social media and the influence of grandmothers that are specific to the rapidly changing urban and peri-urban environments of Vientiane [41] .

## Conclusion

This study shows that a high prevalence of malnutrition among children aged under 24 months persists even in urban settings, and provides a key new insight that uniquely underscores that caregiver behavior and motivation, shaped by information from multiple sources and involving practical skills, differentiates nutritional outcomes in children even when food availability is not a limiting factor. The primary conclusion is that, despite abundant food in these urban, and peri-urban areas, a relatively high proportion of children remain undernourished. The quality of care provided to young children appears central to addressing this issue. A key distinction between caregivers of well-nourished and poorly nourished children is their ability to translate knowledge into practice. Although all parents possessed knowledge about nutritious foods for children, some did not seem to apply this knowledge effectively. Caregivers providing better care accessed information from diverse sources, whereas those with undernourished children relied mainly on health services. These findings suggest that future interventions both in Lao PDR and in other low- and middle-income countries should not only increase knowledge but also enhance communication methods using a range of ways and channels, such as demonstrations and multimedia content. They should promote practical skills for application of nutritional guidance within an interdisciplinary framework for systemic improvement. Furthermore, addressing caregiver motivation and contextual barriers may help to explain more meaningful improvements in child nutritional outcomes. Urban nutrition interventions should adopt a multi-level approach that incorporates caregiver empowerment alongside existing strategies.

## Supporting information

**S1 File. Supplementary tables (S1 to S4_Annex 1 to annex 4_260526_cleaned).**
(DOCX)

## Acknowledgments

We would like to acknowledge the Lao Tropical and Public Health Institute, as well as its donors and collaborators, for their assistance in carrying out the wider study from which the data were derived. Sincere appreciation and particular thanks go out to MCNV Lao PDR and the European Union (EU) for their support.

## Author contributions

**Conceptualization:** Kethmany Ratsavong, Dirk Essink, E. Pamela Wright.

**Data curation:** Kethmany Ratsavong, Dirk Essink.

**Formal analysis:** Kethmany Ratsavong, Somphou Sayasone.

**Investigation:** Kethmany Ratsavong.

**Methodology:** Kethmany Ratsavong, Dirk Essink.

**Resources:** Kethmany Ratsavong, E. Pamela Wright.

**Supervision:** Dirk Essink, E. Pamela Wright, Sengchanh Kounnavong, Jacqueline E. W. Broerse.

**Writing – original draft:** Kethmany Ratsavong.

**Writing – review & editing:** Kethmany Ratsavong, Dirk Essink, E. Pamela Wright, Somphou Sayasone, Sengchanh Kounnavong, Jacqueline E. W. Broerse.

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
