## [Decision Letter · Decision Letter 0]

10 Dec 2025

PONE-D-25-52724Why are some children under 24 months still undernourished in urban and peri-urban Vientiane? A mixed-methods studyPLOS One

Dear Dr. Ratsavong,

Thank you for submitting your manuscript to PLOS ONE. After careful consideration, we feel that it has merit but does not fully meet PLOS ONE’s publication criteria as it currently stands. Therefore, we invite you to submit a revised version of the manuscript that addresses the points raised during the review process.

We look forward to receiving your revised manuscript.

Kind regards,

Satabdi Mitra, M.D(Community Medicine )

Academic Editor

PLOS One

**Journal Requirements:**

“The work was supported by the EU-funded LEARN Project (Number: DCI/SANTI/2014/342-306) and MCNV Lao PDR.”

5. Please expand the acronym “MCNV Lao PDR.” (as indicated in your financial disclosure) so that it states the name of your funders in full.

6. Please note that funding information should not appear in any section or other areas of your manuscript. We will only publish funding information present in the Funding Statement section of the online submission form. Please remove any funding-related text from the manuscript.

Reviewers' comments:

Reviewer's Responses to Questions

**Comments to the Author**

1. Is the manuscript technically sound, and do the data support the conclusions?

Reviewer #1: Yes

Reviewer #2: Partly

2. Has the statistical analysis been performed appropriately and rigorously? 

Reviewer #1: N/A

Reviewer #2: Yes

3. Have the authors made all data underlying the findings in their manuscript fully available?

Reviewer #1: Yes

Reviewer #2: Yes

4. Is the manuscript presented in an intelligible fashion and written in standard English?

Reviewer #1: Yes

Reviewer #2: Yes

5. Review Comments to the Author

Reviewer #1: The research combined a quantitative survey of 333 mother-child pairs with qualitative interviews of 47 caregivers to explore underlying factors. The findings revealed that food insecurity was not the primary issue; instead, the key determinants were caregiver-related behaviors, such as over-reliance on health services for nutrition information, failure to translate knowledge into practice, and the detrimental influence of other caregivers (like grandmothers) whose traditional beliefs often prioritized filling the child's stomach over providing adequate nutritional diversity.

Reviewer #2: This study addresses a critical gap, on why undernutrition persists in urban areas despite food availability along with a Mixed-methods approach that enriches interpretation.

Introduction and methods:

1. The rationale for selecting only one urban and one peri-urban district should be more explicitly justified, as it might affect the generalizability.

2. Please specify, how the data saturation was ensured.

3. Reducing the sample size reduces power of the study; So, did authors ensure if the sample size remained sufficient.

4. Type of sampling not mentioned.

Results and Discussion:

1. Reliability of mothers' self-recorded diet history should be discussed.

2. Potential recall bias and social desirability bias should be acknowledged.

3. Whether inter-examiner reliability measured? Or is the data collected by a single examiner?

4. Qualitative study: Classification bias is not discussed. Coding and software used are not specified.

5. Add a more explicit statement of what new insights this study adds.

6. PLOS authors have the option to publish the peer review history of their article (what does this mean?). If published, this will include your full peer review and any attached files.

Reviewer #1: No

Reviewer #2: No

---

## [Author Response · Author response to Decision Letter 1]

23 Jan 2026

Reviewer #1

Comment Response

Methodology

the research combined a quantitative survey of 333 mother-child pairs with qualitative interviews of 47 caregivers to explore underlying factors. The findings revealed that food insecurity was not the primary issue; instead, the key determinants were caregiver-related behaviors, such as over-reliance on health services for nutrition information, failure to translate knowledge into practice, and the detrimental influence of other caregivers (like grandmothers) whose traditional beliefs often prioritized filling the child's stomach over providing adequate nutritional diversity. Thank for your appreciation of our work.

Reviewer #2

Comment Response

Introduction and methods:

1. The rationale for selecting only one urban and one peri-urban district should be more explicitly justified, as it might affect the generalizability.

The study aimed to explore factors underlying persistently high rates of undernutrition among young children in urban and peri-urban districts of Vientiane Capital in Lao PDR. We randomly selected one district representing the urban area (Saysettha) and one representing the peri-urban area (Pakgeum), located more than 60 km from the city center, as they represent the rapid nutritional transition occurring in Vientiane capital. We acknowledge in the limitations section that, due to time constraints and limited budget, the study was conducted in only two districts, which were selected randomly to represent the two areas. The findings are intended to provide valuable insights applicable to similar urbanizing contexts across Low- and Middle-Income Countries (LMICs).We have added this justification to the Methods section, line 76

2. Please specify, how the data saturation was ensured.

For the qualitative component, we conducted semi-structured interviews with 47 caregivers. Data saturation was monitored during transcription and initial open coding; recruitment was finalized when no new themes related to the study aims emerged from the last three interviews.

3. Reducing the sample size reduces power of the study; So, did authors ensure if the sample size remained sufficient.

We have provided a technical justification in Annex 1. Using the standard Cochrane formula with a conservative prevalence estimate from LSIS III, the required minimum sample was 227. Our final recruitment of 333 exceeds this by over 46%, ensuring the study maintains robust statistical power (>80%) for both the bivariate and multivariate analyses presented in the Results.

4. Type of sampling not mentioned. We conducted a cross-sectional study using sequential exploratory mixed methods. For the quantitative phase, districts were randomly selected, and villages were selected based on the number of children. For the qualitative phase, preliminary quantitative data were used to identify households with either well-nourished or the least well-nourished children (a form of purposive sampling based on criteria outlined in Annex 1.1 and 1.2), and then caregivers in these households were randomly selected for semi-structured interviews.

Results and Discussion:

1. Reliability of mothers' self-recorded diet history should be discussed.

We agree that the reliability of maternal-reported dietary data is a critical consideration. In the revised manuscript, we have added a discussion regarding the reliability of the 7-day food tally and 24-hour recall methods used. To enhance reliability, we employed trained enumerators to conduct face-to-face interviews and used standardized food portion images from the food photobook (line 109) to assist mothers in recalling intake. While we acknowledge the risk of over-reporting 'healthy' foods due to social desirability bias, the 7-day period was chosen specifically to capture a more stable representation of the child's typical diet compared to a single 24-hour recall. We have now addressed this in the 'Limitations' and 'Discussion' sections, line 645.

2. Potential recall bias and social desirability bias should be acknowledged.

We agree and have now explicitly acknowledged these potential biases in the Strengths and Limitations section, line 654. We have discussed how maternal recall may be subject to recall bias regarding exact food frequencies over the 7-day period and how social desirability bias may have influenced caregivers to report practices that align with health recommendations found in the national maternal and child health 'pink' handbooks. We have also detailed the mitigation strategies used, such as non-judgmental interviewing techniques and the use of food photobooks to minimize these effects in line 661.

3. Whether inter-examiner reliability measured? Or is the data collected by a single examiner?

While data were collected by a team of maternal and child health nurses rather than a single examiner, we ensured high inter-examiner reliability through a rigorous standardization protocol. Prior to data collection, all examiners had three days of training (two days on the questionnaire and one on anthropometry); one more day was used to pretest the form. To ensure accuracy in food identification and portion size estimation, a standardized food photobook was used to guide participants during the interviews on how to use the food photo book, the data underwent a dedicated one-day anthropometry orientation focused on standardized SECA measurement techniques. To minimize inter-observer variation, we utilized a 'lead-assistant' model where the nurse performed the measurement and a trained data collector cross-verified the reading before it was electronically recorded. Additionally, equipment was calibrated daily, and senior researchers conducted random spot-checks during the first week of the survey to ensure consistency across the different examiners. These procedures were implemented to ensure that the nutritional status data remained robust and comparable across all study sites. The explanation text as added under the field procedures of the methodology section, line 114.

4. Qualitative study: Classification bias is not discussed. Coding and software used are not specified.

We have revised the Qualitative Data Analysis section to include the specific software and coding procedures used at line 209. To address classification bias, we utilized a 'triangulation' approach where two researchers independently coded the first subset of transcripts. Any discrepancies in theme categorization were discussed until a consensus was reached, ensuring that the findings were grounded in the participants' actual responses rather than researcher bias. We utilized ATLAS.ti version 8 to manage the data and ensure a transparent audit trail of the coding process.

5. Add a more explicit statement of what new insights this study adds. We have strengthened the Conclusion sections to more clearly articulate the novel contributions of this study at line 674. While previous research in Lao PDR has focused on rural poverty among children under five years old, this study provides new insights into the 'urban paradox' of unexpected undernutrition among children under two years of age. It highlights how, in transitioning peri-urban environments, child undernutrition persists not due to a lack of food availability, but due to shifting social dynamics. There is not yet very much reported on this situation which is increasingly common in other low- and middle-income country (LMCIs). This mixed-methods evidence offers a unique perspective on how urban transition creates specific vulnerabilities that require tailored public health interventions.

---

## [Decision Letter · Decision Letter 1]

7 Apr 2026

PONE-D-25-52724R1Why are some children under 24 months still undernourished in urban and peri-urban Vientiane? A mixed-methods studyPLOS One

Dear Dr. Ratsavong,

Thank you for submitting your manuscript to PLOS ONE. After careful consideration, we feel that it has merit but does not fully meet PLOS ONE’s publication criteria as it currently stands. Therefore, we invite you to submit a revised version of the manuscript that addresses the points raised during the review process.

We look forward to receiving your revised manuscript.

Kind regards,

Satabdi Mitra, M.D(Community Medicine )

Academic Editor

PLOS One

Journal Requirements:

Reviewers' comments:

Reviewer's Responses to Questions

**Comments to the Author**

1. If the authors have adequately addressed your comments raised in a previous round of review and you feel that this manuscript is now acceptable for publication, you may indicate that here to bypass the “Comments to the Author” section, enter your conflict of interest statement in the “Confidential to Editor” section, and submit your "Accept" recommendation.

Reviewer #3: (No Response)

Reviewer #4: (No Response)

2. Is the manuscript technically sound, and do the data support the conclusions?

Reviewer #3: Yes

Reviewer #4: Partly

3. Has the statistical analysis been performed appropriately and rigorously? 

Reviewer #3: Yes

Reviewer #4: Yes

4. Have the authors made all data underlying the findings in their manuscript fully available?

Reviewer #3: Yes

Reviewer #4: No

5. Is the manuscript presented in an intelligible fashion and written in standard English?

Reviewer #3: Yes

Reviewer #4: Yes

6. Review Comments to the Author

Reviewer #3: The revised manuscript shows substantial improvement in methodological clarity and analytical depth. The authors have addressed the previous concerns thoughtfully. I appreciate the clearer explanation of sampling procedures. Importatly, the explanation of how potential data reliability concerns were identified and mitigated significantly enhances the methodological transparency of the study. I believe the manuscript in its current form makes a meaningful contribution to the literature on child undernutrition in rapidly urbanizing settings. Overall the manuscript is suitable for publication.

Reviewer #4: This manuscript looks at an important public health issue, which is child undernutrition in urban and peri urban areas. This is a valuable topic because most studies focus more on rural settings. The mixed methods approach is a strength and helps in understanding caregiving practices beyond just food availability.

However, there are several areas that need improvement to make the study clearer and more scientifically strong.

Major comments:

Study design and mixed methods

The authors should explain more clearly why this type of mixed methods approach was chosen. It is also not very clear how the qualitative findings support or add to the quantitative results. There is a long gap between the two phases of data collection. The authors should discuss whether this could affect the findings, especially due to seasonal changes in diet or food access.

Sampling and representativeness

Only one urban and one peri-urban district were included. The authors should explain how well these represent the larger population. The addition of more villages due to initial underestimation is understandable, but it may introduce some selection bias and should be discussed.

Statistical analysis

The statistical methods are generally appropriate, but important details are missing. The authors should explain how variables were selected for the regression analysis. It is also not clear whether multicollinearity and model fit were tested. Converting variables into two groups based on the mean may reduce the quality of the analysis and should be justified.

Definitions and classification

The definition used for well-nourished children does not fully match standard WHO criteria. This needs to be clarified and justified. The method used to define overweight should also be explained more clearly.

Measurement and bias

The study depends partly on mothers recalling and recording dietary information. This can lead to recall bias and reporting bias. Although some steps were taken to reduce this, it should be discussed more clearly. More details are also needed on how consistency between different data collectors was maintained.

Qualitative component

The qualitative part adds value, but more details are needed. The authors should explain how coding was done, how many people were involved, and how consistency was ensured. It would also help to clearly show how qualitative and quantitative findings were combined.

Interpretation of findings

Some conclusions seem stronger than what the data can support, especially since this is a cross-sectional study. The authors should be more careful in suggesting cause and effect.

Data availability

The data availability statement is not very clear. The authors should clearly state whether the full dataset is publicly available or explain any restrictions.

Minor comments:

The manuscript is generally understandable, but the language can be improved for better clarity. Some sentences are long and can be simplified. There are also a few minor grammatical errors that should be corrected. Some parts of the manuscript can be shortened to avoid repetition.

Overall, this is an important study with good potential, but it needs revisions to improve clarity, methods, and interpretation.

7. PLOS authors have the option to publish the peer review history of their article (what does this mean?). If published, this will include your full peer review and any attached files.

Reviewer #3: No

Reviewer #4: **Yes:**Veerabhadra Swamy G S

---

## [Author Response · Author response to Decision Letter 2]

20 May 2026

1.Thank you for your advice. We have updated the manuscript and file naming conventions to strictly adhere to PLOS ONE’s formatting templates.

2.As this study involves research in Lao PDR, we have completed the PLOS Questionnaire on Inclusivity in Global Research and included it as Supporting Information (S5 Checklist). We also added the subsection ‘Inclusivity in global research’ to the Methods section and added the following sentence: “Additional information regarding the ethical, cultural, and scientific considerations specific to inclusivity in global research is included in the Supporting Information (S5 Checklist)” in line 226

3.3. We note that the grant information you provided in the ‘Funding Information’ and ‘Financial Disclosure’ sections do not match.

When you resubmit, please ensure that you provide the correct grant numbers for the awards you received for your study in the ‘Funding Information’ section We have corrected the discrepancy to ensure the grant information is consistent. The funding was supported by the EU-funded Lao Equity through Policy Analysis and Research Networks (LEARN) Program, (Number: DCI/SANTI/2014/342-306) and the Medical Committee Netherlands-Vietnam in Lao PDR (MCNV Lao PDR).

4.The following statement will be included in the cover letter: "The funders had no role in study design, data collection and analysis, decision to publish, or preparation of the manuscript.

5.We included the expanded name of the funder and of MCNV Lao PDR in the cover letter.

6.All funding-related text has been removed from the main body of the manuscript.

7.We have carefully reviewed the reviewer comments for any recommendations to cite specific previously published works. We note that no specific publications were recommended by the reviewers or the editor. However, we have independently updated our reference list to include recent, relevant literature on urban nutrition in Southeast Asia to further strengthen the discussion of our findings.

8.We have conducted a thorough review of the reference list to ensure accuracy and completeness. We cross-checked all cited works against the Retraction Watch database and the NLM PubMed 'retracted publication' filter. We confirm that no retracted papers are cited in this manuscript. Furthermore, we have verified that all references are formatted according to the PLOS ONE 'Vancouver' style, and all URLs/DOIs provided in the references are active and correct. Any minor typographical corrections to the reference list made during this revision have been highlighted in the 'Revised Manuscript with Track Changes' file.

---

## [Editor Report · Decision Letter 2]

24 May 2026

Why are some children under 24 months still undernourished in urban and peri-urban Vientiane? A mixed-methods study

PONE-D-25-52724R2

Dear Dr. Kethmany Ratsavong,

We’re pleased to inform you that your manuscript has been judged scientifically suitable for publication and will be formally accepted for publication once it meets all outstanding technical requirements.

Kind regards,

Satabdi Mitra, M.D(Community Medicine )

Academic Editor

PLOS One
---

## [Editor Report · Acceptance letter]

PONE-D-25-52724R2

PLOS One

Dear Dr. Ratsavong,

I'm pleased to inform you that your manuscript has been deemed suitable for publication in PLOS One. Congratulations! Your manuscript is now being handed over to our production team.

Kind regards,

on behalf of

Dr Satabdi Mitra

Academic Editor

PLOS One